# Links between Immune Cells from the Periphery and the Brain in the Pathogenesis of Epilepsy: A Narrative Review

**DOI:** 10.3390/ijms22094395

**Published:** 2021-04-22

**Authors:** Gaku Yamanaka, Shinichiro Morichi, Tomoko Takamatsu, Yusuke Watanabe, Shinji Suzuki, Yu Ishida, Shingo Oana, Takashi Yamazaki, Fuyuko Takata, Hisashi Kawashima

**Affiliations:** 1Department of Pediatrics and Adolescent Medicine, Tokyo Medical University, Tokyo 160-8402, Japan; s.morichi@gmail.com (S.M.); t-mori@tokyo-med.ac.jp (T.T.); vandersar_0301@yahoo.co.jp (Y.W.); shin.szk@gmail.com (S.S.); ishiyu@tokyo-med.ac.jp (Y.I.); oanas@tokyo-med.ac.jp (S.O.); tyamaz@tokyo-med.ac.jp (T.Y.); 2Department of Pharmaceutical Care and Health Sciences, Faculty of Pharmaceutical Sciences, Fukuoka University, Fukuoka 814-0180, Japan; ftakata@fukuoka-u.ac.jp (F.T.); hisashi@tokyo-med.ac.jp (H.K.)

**Keywords:** epilepsy, cytokine, monocytes, pericytes

## Abstract

Accumulating evidence has demonstrated that the pathogenesis of epilepsy is linked to neuroinflammation and cerebrovascular dysfunction. Peripheral immune cell invasion into the brain, along with these responses, is implicitly involved in epilepsy. This review explored the current literature on the association between the peripheral and central nervous systems in the pathogenesis of epilepsy, and highlights novel research directions for therapeutic interventions targeting these reactions. Previous experimental and human studies have demonstrated the activation of the innate and adaptive immune responses in the brain. The time required for monocytes (responsible for innate immunity) and T cells (involved in acquired immunity) to invade the central nervous system after a seizure varies. Moreover, the time between the leakage associated with blood–brain barrier (BBB) failure and the infiltration of these cells varies. This suggests that cell infiltration is not merely a secondary disruptive event associated with BBB failure, but also a non-disruptive event facilitated by various mediators produced by the neurovascular unit consisting of neurons, perivascular astrocytes, microglia, pericytes, and endothelial cells. Moreover, genetic manipulation has enabled the differentiation between peripheral monocytes and resident microglia, which was previously considered difficult. Thus, the evidence suggests that peripheral monocytes may contribute to the pathogenesis of seizures.

## 1. Introduction

The World Health Organization estimates that at least 50 million individuals are affected by epilepsy worldwide [1]. One-third of these patients develop drug-resistant epilepsy (DRE), which is accompanied by a deterioration in the quality of life (associated with epileptic seizures) and cognitive dysfunction [2]; these factors increase the risk of suicide and sudden unexpected death in epilepsy (SUDEP). SUDEP is the most common cause of epilepsy-related mortality, accounting for 7–17% of deaths in patients with epilepsy and up to 50% in those with refractory epilepsy [3,4]. Epilepsy is among the diseases that are most intractable to treatment, despite the availability of various treatment options.

Previous studies have provided evidence of the activation of the innate and adaptive immune responses in brain tissues using experimental models and human patients with temporal lobe epilepsy (TLE) [5,6,7,8]. Although surface markers can differentiate between the CD4 and CD8 cells involved in the adaptive immune response, the monocytes and microglia that contribute to innate immunity are not well differentiated [9,10]. Monocytes can migrate to various tissues and differentiate into macrophages, and thus invade the brain, where they can differentiate into “microglia-like cells” [9,10]. However, modern genetic manipulation procedures can apparently distinguish between the invading monocytes and resident microglia; the evidence amassed suggests that the peripheral monocytes infiltrating the brain contribute to the pathogenesis of seizures [11,12,13,14].

Current studies have demonstrated that T cells have also been detected in epileptogenic areas of the human brain, even in cases without evidence of an underlying infection or immune disorder [15,16,17,18]. On the other hand, studies have reported that the T-cell lineage is almost undetectable in recent mouse models [19], and that acquired immunity is thought to vary for each species [20].

The evidence accumulated by various studies has proven that the pathogenesis of epilepsy is linked to neuroinflammation and cerebrovascular dysfunction [21,22,23]. The infiltration of the peripheral immune cells into the brain parenchyma was suggested to occur after the disruption of the blood–brain barrier (BBB), which is a known feature of epilepsy [24]. However, studies on experimental allergic encephalomyelitis (EAE) have demonstrated that the destruction of the BBB and the infiltration of the immune cells are two distinct events that occur several days apart [25]. Although BBB destruction leads to plasma protein leakage and ion imbalance, it may not be adequate to induce the invasion of immune cells into the central nervous system (CNS) [25,26,27]. Some of the studies analyzed in this review reported on the existence of a temporal dissociation between BBB-induced leakage and cell infiltration [11,12,14,19]; it is unlikely that cells simply infiltrate the brain secondary to BBB disruption-induced leakage, although this phenomenon could facilitate the release of chemoattractants into the circulation [20]. The series of reactions that lead to the recruitment of peripheral cells and their infiltration into the brain are mediated by various chemical mediators produced by the neurovascular unit (NVU), consisting of neurons, perivascular astrocytes, microglia, pericytes, and endothelial cells. Studies have observed the significant contributions of the C-C motif ligand 2 (CCL2) and interleukin (IL)-1β to these responses [11,12]. The former affects monocyte migration, while the latter possesses neurotoxic and pro-convulsant properties, and both cytokines are implicitly involved in the pathogenesis of epilepsy [21,28].

However, the contribution of the relationship between the peripheral nervous system and CNS to seizures and epilepsy-related pathologies is poorly understood. Therefore, this narrative review sought to explore the current literature on the role of the link between the peripheral nervous system and CNS in the pathogenesis of epilepsy, and highlight novel directions for research into therapeutic interventions for epilepsy that target these reactions.

## 2. Review

### 2.1. Innate Immunity

#### Microglia and Monocytes

Glial cells comprise more than 90% of the cellular component of the human brain and can be categorized into two main populations: the macroglia (astrocytes and oligodendrocytes) and the microglia [29]. Microglial cells are known as the resident macrophages of the CNS and are widely distributed throughout the brain and spinal cord [10]. Microglia are observed before myelopoiesis during the fetal period [10]. The progenitor cells in the yolk sac migrate to the brain via the circulatory system (formed at 8.5–10 days of fetal life) at 7.5 days of fetal life and differentiate into microglia [30]. The microglia are then thought to maintain their numbers, albeit slowly, through self-renewal [31,32]. On the other hand, monocytes originate from the hematopoietic stem cells in the bone marrow, specifically by the gradual and continuous proliferation of the progenitor cells [33]. The monocytes can migrate to various tissues and differentiate into macrophages, which enables them to infiltrate the brain and also differentiate into “microglia-like cells.” Therefore, differentiation between these infiltrating “microglia-like cells” and the true resident microglia is a difficult task [9,10]. Nevertheless, a recent microglial ablation study reported that monocytes can imprint the CNS microenvironment, but remain transcriptionally, epigenetically, and functionally distinct [34]. Studies have also reported that the activated resident microglia and infiltrating monocytes exhibit different morphological and electrophysiological characteristics [14].

Although it is extremely difficult to differentiate between resident microglia and infiltrated monocytes, flow cytometry and genetic manipulation are two of the most common methods that can apparently distinguish between the two cells.

### 2.2. Differentiation between Microglia and Monocytes

#### 2.2.1. Markers of Monocytes

Monocytes can typically be identified based on the expression of CD14, CD16, CD64, and the C-C chemokine receptor type 2 (CCR2) in humans [35,36], and Ly6C, CD43, CD11b, and CCR2 in mice [37,38]. In addition to these markers, CD11b, CD14, CD16, and CD64 are also known to be expressed in the microglia [39,40]. As human and mouse microglia are highly homologous, these markers are expressed in both [41]. Some studies have indicated that CD163 may be a specific marker for monocytes [19,42], although it is also expressed by the microglia [43]. The genetic labeling method has recently become the mainstay for labeling monocytes, although CD169 staining was considered to be a specific marker for monocytes [44,45].

#### 2.2.2. Genetic Modulation Methods

CCR2-red fluorescent protein (RFP) mice [11,12,14] have been used to investigate the role of monocytes in epilepsy. Although CCR2 itself has been suggested to be expressed in several cell types, including the astrocytes and microglia of the CNS [46,47], with the exception of the microglia [48], CCR2-RFP labeling is thought to be specific for investigating the role of monocytes in CNS pathologies [49]. The recent advances in genetic profiling have identified the binding adaptor molecule 1 (IBA1) (a common marker), P2Y12 [45], and transmembrane protein 119 (TMEM119) [50] as markers specific to microglia (Table 1).

### 2.3. How Do Peripheral Monocytes Penetrate the Brain in the Pathogenesis of Epilepsy?

The infiltration of the brain parenchyma by peripheral immune cells may be considered to be a natural secondary sequela of BBB destruction [24]. Prolonged seizures are known to be associated with increased BBB permeability, along with multiple changes in BBB properties [51,52]. However, a time discrepancy exists between BBB disruption and monocyte invasion. Monocyte infiltration is rarely observed until 24 h after the induction of status epilepticus (SE) by kainic acid (KA) [11,12,14], while BBB permeability increases rapidly within 6 h of KA administration, and the BBB is damaged within at least 24 h [52]. The infiltrating monocytes in the KA-induced seizure model were identified in the hippocampal cornu ammonis 3 (CA3) and CA1 in severe seizures, and only in the CA1 in mild seizures; the severity of seizures influences the region of monocytic infiltration [14], but is not likely to influence the time interval between peripheral and central infiltration [14].

Pilocarpine is a cholinergic agonist used as a tool to induce seizures. The peak BBB disruption also occurred at 5 h in the pilocarpine SE mouse model [53], and the period of monocyte infiltration was similar to that reported for the KA-induced SE mouse model [11]. Moreover, the infiltration of the CNS was thought to occur simply due to an increase in the peripheral blood cells induced by the systemic action of pilocarpine-induced SE, but it has been confirmed that pilocarpine does not increase the peripheral blood cells [11].

### 2.4. Disruptive and Non-Disruptive Changes in the BBB

The epileptic model has shown that BBB alterations consist of both disruptive changes, i.e., BBB leakage, and nondisruptive changes at the molecular level [52].

#### 2.4.1. Disruptive Changes in the BBB

The disruptive changes occur within minutes, as shown by Evans Blue staining, and excessive activation causes NVU dysfunction, leading to rapid BBB leakage; these changes include endothelial damage, structural changes in the astrocytes, the destruction of tight junctions, increased vesicular traffic, and the breakdown of the glia limitans [52,54]. The fluorescence method demonstrated that BBB disruption occurs within 5 min of convulsions in rats [55] and 10 min in pigs [56]. However, monocytic infiltration was not observed at this point not only in the epileptic model [11,12,14], but also in the other models [25,26].

The disruption of the BBB in EAE, the animal correlate of multiple sclerosis, was the highest at day 11, as shown by gadolinium-diethylenetriamine pentaacetic acid enhancement; monocyte infiltration was detected at the peak of the disease between days 14 and 17 [25]. A traumatic brain injury model demonstrated that BBB damage was observed between 2 and 12 h post-trauma, and monocytes were observed within the cortical parenchyma at 24 h, which completely filled the cortical lesion site within 72 h of injury [26]. This time discrepancy between BBB disruption and monocytic invasion implies that the rapid alteration of the BBB (mainly leakage) alone may not allow for monocyte infiltration.

#### 2.4.2. Non-Disruptive Changes

The release of cytokines/chemokines and/or enzymes by the astrocytic endfeet, endothelial cells, and pericytes are among the demonstrated non-disruptive changes in the BBB [52]. The release of pro-inflammatory cytokines, such as IL-1β, and proteases, such as matrix metalloproteases, by NVU cells, and the infiltration of leukocytes into the brain, can subsequently lead to disruptive changes, such as the destruction of tight junctions and the extracellular matrix [57,58]. Cellular stress stimulates the release of danger signals within minutes to hours, which subsequently activates the Toll-like receptors on the glial cells, which further activate the inflammatory genes and proteases, causing further BBB damage within hours to days [54].

IL-1β has been identified to be one of the most implicitly involved cytokines in the pathogenesis of epilepsy; IL-1β directly excites neurons via *N*-methyl-d-aspartate receptor activation [59]. This IL-1β-induced increase in neuronal excitability exacerbates neuronal hyperactivity and excitotoxicity, resulting in severe hippocampal degeneration [12]. An elevation in IL-1β mRNA, mainly in the glial cells of the hippocampus, was observed 2 h after SE induction with pilocarpine [60] and KA [61], and 1 to 2 h after heat-induced convulsions in febrile SE models [62]. Tumor necrosis factor (TNF)-α, a well-known pro-inflammatory pleiotropic cytokine also involved in the etiology of epilepsy [21,63], can induce several responses, including proliferation, apoptosis, and inflammation [64]. TNF-α mRNA decreased in the hippocampus 24 h after SE, and significantly increased 5 days after SE, indicating a different behavior to IL-1β [60].

Both IL-1β and TNF-α promote the production of pro-inflammatory chemokines, such as CCL2, CCL20, and the C-X-C motif ligand 2 [65,66], and the potent chemoattractant intercellular adhesion molecule 1 (ICAM)-1 [67,68], which induce immune cell migration, in addition to exacerbating BBB disruption and neuroinflammation. CCL2, which is an especially potent chemoattractant for cells of monocytic lineage [69,70,71], is produced by hyperactive neurons and microglia [12]. The following changes were observed in the mouse model of pilocarpine-induced seizures: CCL2 was expressed at 2 h, peaked at 24 h, and returned to baseline levels after 5 days [60]. The expression of CCL2 also increased 1–3 days after intrahippocampal KA injection [12,72,73,74]. Upregulated CCL2 was also detected in the brain tissue of patients with epilepsy [75,76].

### 2.5. Identification of Peripheral Monocytes in the Brain

The activation of CCL2 triggers the adhesion of monocytes to the inflamed endothelium via β-integrin pathways, and subsequently, the invasion of the brain parenchyma [77]. Recent rigorous studies have highlighted that peripheral monocytes were identified in the brain parenchyma 1 day after pilocarpine- or KA-induced seizures using the genetic modulation method of CCR2-RFP [11,12,14]. Monocytic infiltration peaked 3 days after KA administration in the CA3 region, and declined by the seventh day [11,14].

### 2.6. Potential for Therapy by Controlling Monocytes

Knocking out CCR2, a receptor for CCL2, can virtually abolish the KA-induced upregulation of IL-1β [12]; the activated resident microglia and infiltrated macrophages are both sources of upregulated IL-1β after KA-induced seizures, as evidenced by the confirmation of IL-1β expression in the microglia and monocytes [78]. Another recent study showed similar results concerning the role of invading myeloid cells (including monocytes) after SE, wherein the myeloid cells exhibited higher levels of TNF-α and IL-1β compared to those in the microglia [79]. However, Varvel et al. reported that blocking the entry of monocytes using the CCR2 knockout model mouse did not change the mRNA levels of iNOS, CCL2, TNF-α, or IL-6 in the hippocampal tissues, but reduced IL-1β by 50% (Table 2). The invading monocytes showed a 200-fold elevation in IL-1β mRNA compared to the control microglia; thus, the invading monocytes were found to be the main source of IL-1β [11], although some studies suggest that neurons [12] and astrocytes [80] are the sources of IL-1β. Although the activation of CCR2 induces signal transducer and activator of transcription 3 (STAT3) phosphorylation and IL-1β production, the suppression of STAT3 by WP1066 can inhibit the seizure-induced expression of IL-1β and IBA-1, a microglial maker. However, the contribution of STAT3 to microglial activation and monocyte infiltration could not be strictly distinguished [12]. Interestingly, our recent studies indicated that brain pericytes activate the microglia by releasing IL-6 through the JAK-STAT3 pathways, resulting in BBB breakdown [81,82]. IL-1β is also produced by pericytes [83], and the principal cellular source of IL-1β is still being debated. The evidence accumulated is suggestive of the critical role of STAT3 in CCL2/CCR2-mediated microglial activation and IL-1β production, while the resident microglia, infiltrating monocytes, and brain pericytes may be involved in the anticonvulsant effect of the STAT3 inhibitor.

### 2.7. Association between the Clinical Picture and the Monocytes or Microglia

From the clinical perspective, our recent flow cytometry analysis of pediatric patients with DRE demonstrated higher levels of intracellular cytokines, such as IL-1β, in the peripheral monocytes compared to the controls, which was correlated with the frequency of seizures [84]. The examination of other cells, including T cells, and natural killer and natural killer T cells, and plasma cytokines levels, including CCL2, TNF-α, and IL-6, did not yield such results [84]. Moreover, the analysis of the brain tissue of pediatric patients with DRE revealed that the seizure frequency was correlated with the number of infiltrating peripherally monocytes, but not that of the microglia [16].

Contrary to these results, other studies have found a positive correlation between the degree of microgliosis and the severity of neuronal death and seizures [14,85], which suggests that the resident microglia are primarily responsible for the pathogenesis of epilepsy, instead of the peripheral monocytes. Monocytes contribute to the inflammatory milieu in the brain simply by virtue of their infiltration, even in the absence of proinflammatory cytokine induction, whereas the induced expression of cytokines contributes to neuroinflammation in the microglia [11]. However, the current study demonstrated that non-inflammatory changes in the microglia disrupted the homeostasis of the CNS, and caused a prominent decrease in synaptic density, while the microglia themselves infiltrated the hippocampal pyramidal layer. These series of reactions lead to neuronal degeneration and massive astrocytic proliferation, resulting in the development of severe early onset spontaneous recurrent seizures in mice [86]. Although research has yet to clarify whether the peripheral monocytes or resident microglia are the main triggers of epilepsy, the results of our review suggest that the resident microglia and peripheral monocytes may be closely involved in the pathogenesis of refractory epilepsy in children.

### 2.8. Adaptive Immunity

#### 2.8.1. Experimental Evidence of the Invasion of Peripherally Adapted Immune Cells into the Brain

There also exists mounting experimental evidence of the involvement of the adaptive immune response in epilepsy and seizure disorders [5,7,15,58,79,87,88,89]. Unbiased quantitative analysis revealed a gradual accumulation of CD3-positive, mainly CD8-positive T cells (60–75%) in KA-treated mice, which attained peak levels 2 weeks after injection and persisted at 4 weeks. Initially, the CD3-positive T cells were detected within the blood vessels but were found mainly within the neuropils at 2–4 weeks [7]. In the case of electrically induced seizures, CD4-positive and CD8-positive T cells, as well as CD45R-positive B cells, appeared in brain parenchyma 24 h after a maximal seizure and reached peak levels at 48 h, but were no longer detected at 7 days. The CD4-positive T cells and CD45R-positive B cells were preferentially found in the neocortex compared to the hippocampus, whereas the CD8-positive T cells were preferentially found in the hippocampus 24 h after a maximal seizure [8]. Another recent study reported that CD4-positive cells were observed 96 h after SE [79]. Thus, the timing of T-cell infiltration varies widely according to different studies. These discrepancies may be attributed to the experimental method and type of mice used, partially due to the fact that CD8- and CD4-positive T cells infiltrate the brain parenchyma via different mechanisms [90,91]. However, a recent study analyzed the pilocarpine-induced SE model and demonstrated that CD3-positive cells could not be detected at any time-point during epileptogenesis in experimental epileptic mouse models [19].

#### 2.8.2. Association between the Clinical Picture and Cells of the T Lineage

The data accumulated by studies on CD8 infiltration into the brain tissue of human patients with TLE are incontrovertible [7,15,58,88,89,92]. The infiltration of acquired immune system cells, including CD8-positive or regulator T (T reg) cells, has also been found in the tissues of patients with focal cortical dysplasia [16,93]. Although some studies also demonstrated the infiltration of CD4-positive T cells [15,94], their evidence was limited.

The greater infiltration of CD8-positive cytotoxic T lymphocytes (CTLs) than that of the CD4-positive or CD68-positive microglia/macrophages in certain CA1 regions of the hippocampus showed a greater positive correlation with neuronal loss [15]. Besides the role of CD68-positive microglia/macrophages, this finding is consistent with the conclusion of previous studies that neuronal loss was significantly greater in the CA1 region than in the other sub-hippocampal regions in hippocampal sclerosis [95], and that CD8-positive CTL infiltration was the main causative factor [7,15,58,88,89,92]. While a recent study demonstrated that IL-17-producing γδ T lymphocytes are concentrated in the epileptogenic zone, and their numbers were positively correlated with seizure severity, the number of brain-infiltrating T reg cells was inversely correlated with disease severity [16].

Neurodegeneration and spontaneous recurrent seizures were markedly exacerbated after KA treatment in recombinant activating gene 1 (RAG1) knockout mice that lacked T and B cells [7]. Both γδ T cell- and IL-17RA-deficient mice and recipients of T reg cells have suppressed seizure activity, whereas inactivating/depleting the T reg cells with either anti-CD25 antibodies [96,97] or thymidine injection [98] exacerbates seizure severity. The innate immune system responds more rapidly to seizure events compared to the acquired immune system, and some researchers argue that the former plays a prominent role in the development of epilepsy [88,92]. Although the activation of the adaptive immune response may depend on the type of epilepsy, animal model, species, or neuropathology [19], these findings indicate that the acquired immune system is also involved in the pathogenesis of epilepsy.

### 2.9. Role of Pericytes in the Link between Peripheral Immune Cells and the Brain

Pericytes provide physical support to the BBB and play an integral role in CNS homeostasis and BBB function [99]. They can regulate the migration of leukocytes across the brain microvascular endothelial cell barrier [100,101,102]. Pericytes can secrete chemokines, including CCL2, and help recruit peripheral immune cells, including monocytes, B and T cells, and neutrophils to the CNS parenchyma via the upregulation of ICAM-1 and vascular cell adhesion molecule 1 on the endothelium [103,104,105,106,107]. Recent studies conducted by us and other researchers have demonstrated that brain pericytes respond to inflammatory signals, such as circulating cytokines, IL-1β, and TNF-α, and convey this information to the surrounding cells through chemokine and cytokine secretions [83,105,106,107,108,109]. Moreover, our recent studies found that pericytes may act as sensors for the inflammatory response in the CNS, as they react more intensely to TNF-a and proinflammatory cytokines than other cell types (e.g., brain endothelial cells or microglia), and release MMP-9 (impairing BBB function) and IL-6 (activating microglia) [81,106,110,111]. Recent experiences highlight the substantive role of pericytes in the pathogenesis of epilepsy [23,112,113,114,115,116]. Cerebrovascular pericytes undergo redistribution and remodeling, potentially contributing to BBB permeability, and inflammatory cytokines including IL-1β, TNFα, and IL-6 are deeply involved in the pathogenesis of pericyte-mediated epilepsy [117].

In addition, it was demonstrated that, apart from IL-6, these inflammatory cytokines induced IL-8 and MMP-9 release from brain pericytes, and then reduced neutrophil adhesion to brain pericytes and facilitated neutrophil transmigration across the BBB [118]. Based on these data, the possibility that brain pericytes act as sensors of inflammatory stimuli and effectors of inflammatory stimuli through MMP-9 and IL-8 release, and therefore have a role in peripheral immune cell transmigration to the brain parenchyma and seizure development, should be considered. In fact, pericyte regulation leads to seizure suppression, as demonstrated in vivo [115] and in vitro [116].

Pericyte regulation leads to the suppression of seizures, as demonstrated in vivo [116] and in vitro [115]. These results underline the potential use of pericytes as a therapeutic target for seizure disorders. The role of pericytes in maintaining BBB integrity and recruiting leukocytes suggests that they may be involved in the pathogenesis of peripheral immune cells in epilepsy, but so far, there is no evidence to confirm this hypothesis. In the pathogenesis of epilepsy, pericytes take on a phenotype that is neither pro- nor anti-inflammatory only [118]. Pericyte suppression may not be sufficient to improve the treatment of epilepsy, and may need to be combined with various therapies for a tailored treatment for affected children.

Prolonged seizures can cause disruptive changes in the BBB; leakage can occur within minutes (5 min of convulsions in rats [55] and 10 min in pigs [56]) and can result in the exposure of brain tissue to proteins from the blood vessels. This is followed by non-disruptive changes in the BBB with the release of cytokines and other substances. The upregulation of IL-1β, mainly in the glial cells of the hippocampus, was detected about 2 h after SE [60,61]. IL-1β stimulates the production of CCL2 [66] and the potent chemoattractant ICAM-1 [68] by the microglia, vascular endothelial cells, pericytes and neurons; the activation of CCL2 triggers the adhesion of monocytes to the inflamed endothelium, followed by infiltration into the brain parenchyma [77]. Peripheral monocytes are identified in the brain parenchyma 1 day after the seizure, peaking 3 days later, and decrease by the seventh day [11,12,14].

The components of adaptive immunity, i.e., the CD4+ and CD8+ T cells, appear in brain tissue 24–48 h after convulsions and disappear after 7 days [8]. However, some studies have reported that CD4 cells were observed 96 h after SE [79], and that the T-cell lineage was not identified in the brain tissue after seizures [19].

BBB: blood–brain barrier; CCL2: chemokines chemokine C-C motif ligand 2; SE: status epilepticus; ICAM-1 intercellular adhesion molecule 1.

## 3. Conclusions

In this review, we found evidence that the peripheral immune cells, especially monocytes, may be involved in the pathogenesis of epilepsy via invasion of the CNS. Genetic manipulation has enabled the differentiation between peripheral monocytes and resident microglia, which was historically the greatest obstacle in examining the mechanism by which the peripheral monocytes invade the CNS and participate in the pathogenesis of epilepsy [11,14,78]. However, the issue of monocyte infiltration remains challenging because this method cannot be used in humans. With respect to the time course of innate immune responses in mouse models of epilepsy, recent reports have consistently shown a relatively acute innate immune response (1–3 days) that disappears in 7–14 days [11,12,14]. As for the examination of acquired immunity, CD8 seems to play a major role in humans [7,15,58,88,89,92], but not in mice, with a wide infiltration time scale [7,8,19,79] (Figure 1, Table 3). Despite the general unfeasibility of comparisons between mouse and human studies, there is no denying that peripheral immune cells are involved in the pathogenesis of epilepsy.

Notably, the peripheral monocytes and resident microglia were also found to differ considerably with respect to their morphological, dynamic, and electrophysiological properties [14]: the invading monocytes have distinctly higher levels of IL-1β mRNA compared to the microglia [11]. Interestingly, one study has also suggested that monocytic infiltration triggers the activation of endogenous microglia after seizure induction [12], and that suppressing monocytic infiltration may facilitate the regulation of the resident microglia. In either case, seizures are suppressed by inhibiting the entry of monocytes into the brain by CCR2 knockout [11,69] and CCR2 antagonists [119,120], while the neuroprotective effects of monocytes have also been postulated [121,122]. Moreover, the knockout of T and B cells [7] or the suppression of T reg cells leads to a decline in seizure activity [96,97,98]. Long-term evaluations, including those of cognitive function, of the effects of suppressing the invasion of the immune system into the brain are essential in future experimental models.

Currently, the use of natalizumab (Tysabri) [123] and fingolimod (Gilenya) [124], FDA-approved drugs that prevent the migration of white blood cells into the brain, are recommended for the treatment of relapsing remitting multiple sclerosis (MS). The application to epilepsy treatment is still in its experimental stages, but there are potential natalizumab [125,126] and fingolimod [127,128] treatments. Natalizumab, a humanized monoclonal antibody, is an α4 integrin antagonist of an agents class known as selective adhesion-molecule inhibitors [129] that acts by inhibiting immune cell migration across the BBB [130]. Progressive multifocal leukoencephalopathy (PML) develops as an adverse effect [131], while intractable epilepsy associated with PML has also been reported [132]. Fingolimod is a substrate of sphingosine kinases that binds to sphingosine-1-phosphate receptors [133] and leads to immunomodulation by lymphocyte sequestration, reducing the numbers of T and B cells in circulation [134]. This agent is well tolerated, but may cause hepatic damage, infection, bradycardia, and rarely, leukoencephalopathy [135].

Another alternative could be the use of stem cell therapies. Human pluripotent stem cells, derived from brain pericyte-like cells, can result in the strengthening of the BBB and the reduction of transcytosis [136]. All of the above methods could be applied to develop new strategies to selectively and specifically target pericytes in epilepsy.

Given the scarcity of knowledge on the role of the link between the peripheral nervous system and CNS in seizures and epilepsy-related pathologies, further studies are warranted to investigate these reactions as potential therapeutic targets for epilepsy.

## Figures and Tables

**Figure 1 ijms-22-04395-f001:**
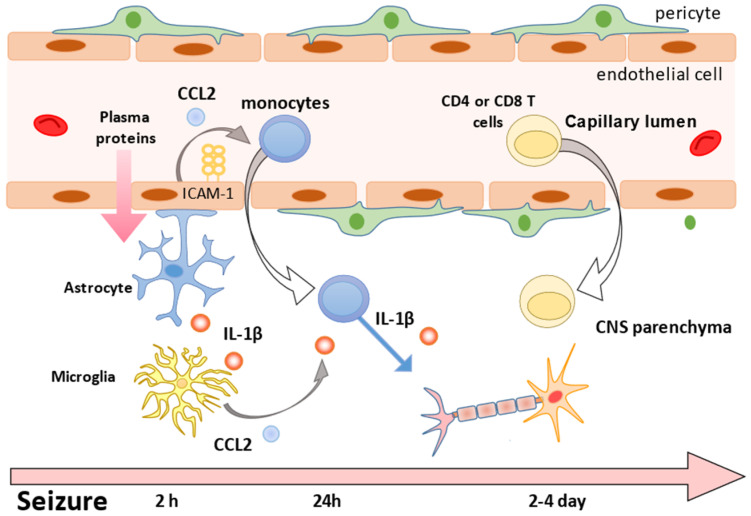
Representation of associations between the peripheral immune cells and brain in experimental models investigating epileptogenesis.

**Table 1 ijms-22-04395-t001:** Genetic marker of monocytes and microglia.

Monocytes	Microglia
CCR2-red fluorescent protein (RFP) [11,12,14]	Identified the binding adaptor molecule 1 (IBA1) P2Y12 [45] Transmembrane protein 119 (TMEM119) [50]

**Table 2 ijms-22-04395-t002:** Immune profiles in representative mouse models.

	Mouse Model	Genetic Mouse Model	mRNA Upregulation in Hippocampus	Significant Points
Varvel et al., 2016 [11]	Pilocarpine Kainic acid	CCR2 ^RFP/+^	IL-1β, IL-6 TNF-α, CCL2, iNOS	Blocking monocyte entry using the CCR2 KO model mouse did not alter iNOS, CCL2, TNF-α, or IL-6 mRNA levels in hippocampal tissues, but reduced IL-1β by 50%.
Tian et al. 2017 [12]	Kainic acid	CX3CR1 ^GFP/+^:CCR2 ^RFP/+^	IL-1 a, IL-1β, IL-1RA, CCL2, CCL3, CCL5, CCL12, CXCL10	Knocking out CCR2R can virtually abolish KA-induced IL-1β upregulation

IL: interleukin, TNF-α: tumor necrosis factor-α, CCL: C-C motif ligand, iNOS: inducible nitric oxide synthase, RA: receptor antagonist CXCL1: C-X-C motif chemokine ligand.

**Table 3 ijms-22-04395-t003:** Time course of innate and adaptive immune responses in experimental models of epilepsy.

	Monocytes	CD4+	CD8+
Kainic acid-treated mice	Appears from day 1, peaks in 3 days, disappears in 7–14 days [11,12,14]	Initially appears within blood vessels but mainly within the neuropils at 2–4 weeks [7]	Appears, peaks 2 weeks after injection and persists at 4 weeks [7]
Pilocarpine-treated		Appears after 96 h [79]	Not described
Electrical stimulation	Not described	Appears after 24–48 h and disappears after 7 days [8]	
		Not identified [19]	

## Data Availability

The datasets generated and/or analyzed during the current study are available at the PubMed database repository (https://pubmed.ncbi.nlm.nih.gov/, accessed on 1 March 2021).

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
