# Peer review of "Links between Immune Cells from the Periphery and the Brain in the Pathogenesis of Epilepsy: A Narrative Review"

_ijms, 2021, doi:10.3390/ijms22094395_

Round 1

Reviewer 1 Report

This is a well thought review, full of details however, to provide clarity a few suggestions

Table: Different type of insults where monocyte invasion yes/no documented, timeframe.

Table: Immune profile in each model and correlation with monocytes.

Specify human vs mouse data- side by side. Similar/different.

Better define markers of microglia vs monocytes (human/mouse)

Better define timeframe innate vs adaptive immune response in epilepsy

Define pericyte role as a sensor allow/not allow passage.

Petit mal vs major seizure - do all lead to monocyte invasion and whether medication that reduce intensity can prevent, what is the threshold?

Author Response

Journal IJMS (ISSN 1422-0067)

Manuscript ID ijms-1177509

Type Review

Number of Pages 18

Title: Links between immune cells from the periphery and the brain in the pathogenesis of epilepsy: A narrative review

To the Reviewer 1

We appreciate the time and effort you have dedicated to providing this insightful feedback. We have revised our manuscript in accordance with your comments, as much as possible. However, we have limited time and may not have been able to make the corrections you mentioned. All suggested revisions, as well as additional ones to improve the language of the manuscript, are indicated in underline and yellow text. We sincerely hope that, with these revisions, our manuscript will be suitable for publication in your esteemed journal.

Reviewer: 1
Comments to the authors

This is a well thought review, full of details however, to provide clarity a few suggestions

Comment 1

Table: Different type of insults where monocyte invasion yes/no documented, timeframe.

Comment 5; Better define timeframe innate vs adaptive immune response in epilepsy

Response to comments 1 and 5

I appreciate your valuable comments. In the reports differentiating monocytes and microglia by genetic methods, the time course of monocyte invasion seems consistent. I created a new table like the one below, however, the information available is limited. 

genetic mouse model

24h

3 day

7-14d

Varvel et al, 2016

KA

CCR2-RFP/+

+

+++

+

Pilocarpine

+

+++

+

Tian et al, 2017

Kainic acid

CX3CR1CreER/+:

R26tdTomato/+

+

+++

+

Feng et al, 2019

Kainic acid

CX3CR1GFP/+ CCR2RFP/+

+

+++

+

As you pointed out in comment 5 (Better define timeframe innate vs adaptive immune response in epilepsy), I created a new table with information on the time axis of innate and adaptive immune responses in epilepsy, which we can include in the manuscript if you consider it appropriate.

Table 2 Time course of innate and adaptive immune responses in experimental models of epilepsy

Monocytes

CD4⁺

CD8⁺

Kainic acid -treated mice

Appears from day 1, peaks in 3 days, disappears in 7-14 days [11, 12, 14]

Initially appears within blood vessels but mainly within neuropils at 2–4 weeks [7]

Peak levels 2 weeks after injection persisting at 4 weeks [7]

Pilocarpine-treated

Appears after 96 h[79]

Not described

Electrical stimulation

Not described

Appears after 24-48 h and disappears after 7 days [8]

Not identified [19]

Comment 2

Table: Immune profile in each model and correlation with monocytes.

Response

The immune profile in each model is very interesting. Again, only a few reports established central monocyte entry by genetic methods, and only two mention the immune profile Varvel et al 2016, Tian et al 2017{Varvel, 2016 #558;Tian, 2017 #1587}. Further, there is no study on the correlation between monocytes and cytokines invading the central nervous system.  We described changes in cytokines by blocking monocytes as a significant point. And created the table below.

Table 1 Immune profiles in representative mouse models

Mouse model

genetic mouse model

mRNA upregulation in hippocampus

Significant points

Varvel et al, 2016 [11]

Pilocarpine Kainic acid

CCR2RFP/+

IL-1β, IL-6 TNF-α, CCL2, iNOS

blocking monocyte entry in the CCR2 KO model mouse did not change mRNA levels of iNOS, CCL2, TNF-α, & IL-6 in hippocampal tissues, but reduced IL-1β by 50%.

Tian et al 2017[12]

Kainic acid

CX3CR1GFP/+

:CCR2RFP/+

IL-1 a, IL-1β IL-1ra CCL2, CCL3, CCL5, CCL12, CXCL10

Knocking out CCR2, can virtually abolish KA-induced upregulation of IL-1β

IL: interleukin, TNF-α: Tumor necrosis factor-α, CCL: C-C motif ligand, iNOS: inducible nitric oxide synthase, RA: receptor antagonist CXCL1: C-X-C motif chemokine ligand

Vinet et al. also examined myeloid cells that invaded the central nervous system, but not strictly monocytes. Therefore, Line 222 has also been corrected as follows.

Another recent study showed similar results on the role of invading myeloid cells (including monocytes) after SE, wherein myeloid cells exhibited higher levels of tumor necrosis factor (TNF)-α and IL-1β compared to those in the microglia[79],

Comment 3

Specify human vs mouse data- side by side. Similar/different.

Response

We also believe that the comparison between human and mouse experiments is very important. Broekaart et al. (Epilepsia. 2018;59:1931-1944) is an excellent paper that examines innate and acquired immunity in humans and mice. However, the manuscript is limited in terms of the monocytes infiltrating the central nervous system because it uses CD163 as a specific marker for monocytes, when CD163 is also expressed in microglia. Since it is currently impossible to apply the aforementioned genetic methods to humans, it is very difficult to study the central invasion of monocytes in human tissues.

 On the other hand, acquired immunity has been studied in humans as described in 2.8.2. but not the time axis, and there is a limitation to studying it in mice and humans. CD8 infiltration is the main feature in humans, but no such tendency has been observed in mice. However, human and mouse studies both show that at least peripheral cells seem to be involved in the pathogenesis of epilepsy.

We included these results in the Conclusion (Lines 392-399).

“Genetic manipulation has enabled the differentiation between peripheral monocytes and resident microglia, which was historically the greatest obstacle in examining the mechanism by which the peripheral monocytes invade the CNS and participate in the pathogenesis of epilepsy [11, 14, 78]. However, the issue of monocyte infiltration remains challenging because this method cannot be used in humans. With respect to the time course of innate immune responses in mouse models of epilepsy, recent reports have consistently shown a relatively acute innate immune response (1-3 days) which disappears in 7-14 days [11, 12, 14]. As for the examination of acquired immunity, CD8 seems to play a major role in humans [7, 15, 58, 88, 89, 92], but not in mice, and a wide range infiltration time range [7, 8, 19, 79]. Despite the general unfeasibility of comparisons between mouse and human studies, there is no denying that peripheral immune cells are involved in the pathogenesis of epilepsy.”

Comment 4 Better define markers of microglia vs monocytes (human/mouse)

Response

As you pointed out, the expression is not clear, so I added the following text (Line 116)

In addition to these markers, CD11b, CD14, CD16, and CD64 are also known to be expressed in the microglia [39, 40]. Because human and mouse microglia are highly homologous, these markers are expressed in both [41].

Comment 5

Better define timeframe innate vs adaptive immune response in epilepsy

Response

As mentioned in our first response, I have created a new table and added the following comments to the conclusion. 

With respect to the time course of innate immune responses in mouse models of epilepsy, recent reports have consistently shown that the innate immune response is relatively acute (1-3 days) and disappears in 7-14 days [11, 12, 14]. As for the examination of acquired immunity, CD8 seems to play a major role in humans [7, 15, 58, 88, 89, 92], but not in mice, and a wide infiltration time scale [7, 8, 19, 79].

Comment 6 

Define pericyte role as a sensor allow/not allow passage.

Response

We sincerely appreciate your recommendations but the role of pericytes in the pathogenesis of epilepsy is still unknown. In the pathogenesis of epilepsy, it has been suggested that pericytes not only reduce inflammation, but also take on an anti-inflammatory phenotype (Rustenhoven, J Neuroinflammation. 2016;13:379. The following text has been amended in the manuscript.

(Lines 335-339)

Moreover, our recent studies found that pericytes may act as sensors for the inflammatory response in the CNS, as they react more intensely to TNF-a, and proinflammatory cytokines than other cell types (e.g., brain endothelial cells or microglia), and release MMP-9 impairing BBB function and IL-6 activating microglia [81, 106, 110, 111].

(Lines 344-351)

In addition, it was demonstrated that, apart from IL-6, these inflammatory cytokines induced IL-8 and MMP-9 release from brain pericytes and then decreased neutrophil adhesion to brain pericytes and facilitated neutrophil transmigration across the BBB [118]. Based on these data, the possibility that brain pericytes act as sensors of inflammatory stimuli and effectors of inflammatory stimuli through MMP-9 and IL-8 release, and therefore have a role in peripheral immune cell transmigration to the brain parenchyma and seizure development, should be considered. In fact, pericyte regulation leads to seizure suppression, as demonstrated in vivo [115] and in vitro [116].

(Lines 358-361)

In the pathogenesis of epilepsy, pericytes take on a phenotype that is neither pro- nor anti-inflammatory only [119]. Pericyte suppression may not be sufficient to improve the treatment of epilepsy and need to be combined with various therapies for a tailored treatment for affected children.

Comment 7

Petit mal vs major seizure - do all lead to monocyte invasion and whether medication that reduce intensity can prevent, what is the threshold?

Response

Thank you for this very interesting question.

As you pointed out, and as we mentioned in Line 147, the severity of the seizures seems to affect monocyte invasion. Infiltrating monocytes in the KA-induced seizure model were identified in the hippocampal CA3 and CA1 in severe seizures, but only in the CA1 in mild seizures: the severity of seizures influences the region of monocytic infiltration (Feng, Glia. 2019;67:1434–1448). This is consistent with the notion that the degree of microgliosis directly correlates with seizure activity (Kim et al., 2015 Journal of Immunology, 195, 3345–3354.).

There seems to be a genetic animal model of petit mal absence, but to our knowledge, there are no reports of monocyte invasion using it. Therefore, we cannot make a comparison between petit mal and major seizure.

We believe it possible to inhibit monocyte invasion and microgliosis by suppressing seizures, but as far as we could find, there are no reports on this threshold. However, we would like to consider this interesting point in the future.

Reviewer 2 Report

In general, the manuscript is interesting and it reads well. It presents recent data showing potential links between epilepsy and immune cells functioning. I have a few concerns regarding its content, which need to be clarified before potential publication:

  1. As stated in the Abstract section, one of the aims of this review is 'to highlight novel research directions for therapeutic interventions targeting these reactions'. Unfortunately, while the potential role of immune cells in the pathogenesis of epilepsy is very comprehensively described, these potential therapies are not. I would ask the authors to add some information about how the described 'immune drug targets' for epilepsy might influence the search for novel therapies for epilepsy (with some examples of tested  compounds and drug candidates).
  2. Section 2.3 - the statement about pilocarpine  must be re-written. Pilocarpine cannot be named a chemical inhibitor. If so, it is a muscarinic receptor agonist, a cholinergic system agonist, etc.
  3. The role of IL-1beta is thoroughly described as one of factors underlying epilepsy, but little is mentioned about other important monocyte-derived pro-inflammatory cytokines, e.g. TNFalpha. The authors are asked to add some data about the role of this cytokine in neuroinflammation in the course of epilepsy (e.g. Patel et al., eNeuro, 2017 and many others).
  4. Conclusion section: as stated above, potential therapies for epilepsy, based on targeting peripheral immune cells invading the CNS are not sufficiently described. In this context the authors focused (in the section Conclusions) on natalizumab and fingolimod, however they did not state that these drugs are registered for multiple sclerosis and their use in epilepsy is totally experimental. Also this issue requires further explanation, as there are also reports showing that natalizumab induces refractory temporal lobe epilepsy (e.g. Abkur et al., Mult Scler Relat Disord, 2018). Please comment more on these (and other) potential therapies (fingolimod and natalizumab) for epilepsy (mechanisms, benefits and risk factors).

Author Response

Journal IJMS (ISSN 1422-0067)

Manuscript ID ijms-1177509

Type Review

Number of Pages 18

Title: Links between immune cells from the periphery and the brain in the pathogenesis of epilepsy: A narrative review

To the Reviewer 2

We appreciate the time and effort that you have dedicated to provide your insightful feedback. We have revised our manuscript according to your comments, as much as possible. All suggested revisions, as well as additional ones to improve the language of the manuscript, are indicated in underline and yellow text. We sincerely hope that, with these revisions, our manuscript will be suitable for publication in your esteemed journal.

Reviewer: 2
Comments to the authors

In general, the manuscript is interesting and it reads well. It presents recent data showing potential links between epilepsy and immune cells functioning. I have a few concerns regarding its content, which need to be clarified before potential publication:

Comment 1

As stated in the Abstract section, one of the aims of this review is 'to highlight novel research directions for therapeutic interventions targeting these reactions'. Unfortunately, while the potential role of immune cells in the pathogenesis of epilepsy is very comprehensively described, these potential therapies are not. I would ask the authors to add some information about how the described 'immune drug targets' for epilepsy might influence the search for novel therapies for epilepsy (with some examples of tested compounds and drug candidates).

Response

Thank you very much for your meaningful remarks.

We made the following correction according to Comment 4 (fingolimod and natalizumab) as well. We also mentioned IPS cells as a possibility to block monocyte invasion, despite it being more experimental.

Currently, the use of natalizumab (Tysabri) {Planas, 2014 #1970} and fingolimod (Gilenya) {Kappos, 2015 #1971}, FDA-approved drugs that prevent the migration of white blood cells to the brain, as potential treatments for pediatric epilepsy is problematic due to their potential for serious adverse effects. Given the scarcity of knowledge on the role of the link between the peripheral nervous system and CNS in seizures and epilepsy-related pathologies, further studies are warranted to investigate these reactions as potential therapeutic targets for epilepsy.

(Lines 413-429)

Currently, natalizumab (Tysabri) [124] and fingolimod (Gilenya) use [125], FDA-approved drugs that prevent migration of white blood cells to the brain, are recommended for treatment of relapsing remitting multiple sclerosis (MS). The application to epilepsy treatment is still in experimental stages, but there are potential natalizumab [126, 127] and fingolimod [128, 129] treatments. Natalizumab, a humanized monoclonal antibody, is an α4 integrin antagonist of an agents class known as selective adhesion-molecule inhibitors [130] that acts by inhibiting immune cell migration across the BBL [131]. Progressive multifocal leukoencephalopathy (PML) develops as an adverse effect [132], while intractable epilepsy associated with PML has also been reported [133]. Fingolimod is a substrate of sphingosine kinases that binds to sphingosine-1-phosphate receptors [134] and lead to immunomodulation by lymphocyte sequestration, reducing the numbers of T and B cells in circulation [135]. This agent is well tolerated, but may rarely cause hepatic damage, infection, bradycardia, and rarely leukoencephalopathy [136].

Another alternative could be the use of stem cell therapies. Human pluripotent stem cell-derived from brain pericyte-like cells can result in strengthening of the BBB and reduction of transcytosis [137]. All of the above methods could be applied to develop new strategies to selectively and specifically target pericytes in epilepsy.

Comment 2

Section 2.3 - the statement about pilocarpine must be re-written. Pilocarpine cannot be named a chemical inhibitor. If so, it is a muscarinic receptor agonist, a cholinergic system agonist, etc.

Response

We agree with your comment, and the following correction accordingly.

Pilocarpine is another chemical inhibitor

 (Line 145)

Pilocarpine is a muscarinic receptor agonist

Comment 3

The role of IL-1beta is thoroughly described as one of factors underlying epilepsy, but little is mentioned about other important monocyte-derived pro-inflammatory cytokines, e.g. TNF-α. The authors are asked to add some data about the role of this cytokine in neuroinflammation in the course of epilepsy (e.g. Patel et al., eNeuro, 2017 and many others).

Response

Thank you for your important remarks. Various cytokines have been recognized as potentially related with the pathogenesis of epilepsy. Indeed, TNFα is also an important monocyte-derived cytokine, which induces chemokines such as CCL2 and ICAM-1 (Sheng J et al. 2005 Dec;78(6):1233-41, Osborn et al. Cell. 1989 Dec 22;59(6):1203-11.; Arisi et al. Journal of Neuroinflammation (2015) 12:129). We made the following changes (Lines 192-199).

Tumor necrosis factor (TNF)-α, a well-known pro-inflammatory pleiotropic cytokine also involved in the etiology of epilepsy [21, 63], can induce several responses including proliferation, apoptosis, and inflammation [64]. TNF-α mRNA decreased in the hippocampus 24 hours after SE, and significantly increased 5 days after SE, indicating a different behavior from IL-1β [60].

Both IL-1β and TNF-α promote the production of pro-inflammatory chemokines, such as CCL2, CCL20, and the C-X-C motif ligand 2 [65, 66], and the potent chemoattractant intercellular adhesion molecule 1 (ICAM)-1 [67, 68], in addition to exacerbating BBB disruption and neuroinflammation.

Comment 4

Conclusion section: as stated above, potential therapies for epilepsy, based on targeting peripheral immune cells invading the CNS are not sufficiently described. In this context the authors focused (in the section Conclusions) on natalizumab and fingolimod, however they did not state that these drugs are registered for multiple sclerosis and their use in epilepsy is totally experimental. Also this issue requires further explanation, as there are also reports showing that natalizumab induces refractory temporal lobe epilepsy (e.g. Abkur et al., Mult Scler Relat Disord, 2018). Please comment more on these (and other) potential therapies (fingolimod and natalizumab) for epilepsy (mechanisms, benefits and risk factors).

Response

This point has already been discussed above. We would like to thank you for your valuable feedback.

Round 2

Reviewer 1 Report

The manuscript was improved by introducing the requested specifications and tables. 

However, the genetic background table is not in the manuscript although it is part of the comments of the paper- please introduce. 

Is there a supplement?

Author Response

Journal IJMS (ISSN 1422-0067)

Manuscript ID ijms-1177509

Type Review

Number of Pages 18

Title: Links between immune cells from the periphery and the brain in the pathogenesis of epilepsy: A narrative review

Reviewer comment: 1
The manuscript was improved by introducing the requested specifications and tables.

However, the genetic background table is not in the manuscript although it is part of the comments of the paper- please introduce.

Comments to Reviewer

We appreciate the time and effort you have dedicated to providing this insightful feedback.  All suggested revisions, as well as additional ones to improve the language of the manuscript, are indicated in underline and yellow text. We have added Table 1 as shown below. Thank you for your valuable comments (Line 133).

Table 1 Genetic marker of Monocytes and Microglia

Monocytes

Microglia

CCR2-red fluorescent protein (RFP) [11, 12, 14]

Identified the binding adaptor molecule 1 (IBA1) P2Y12 [45]                     Transmembrane protein 119 (TMEM119) [50]

Reviewer 2 Report

The authors addressed almost all my concerns but please re-write the description of pilocarpine (line 147). Pilocarpine is a cholinergic agonist used as a tool to induce seizures.

Author Response

Journal IJMS (ISSN 1422-0067)

Manuscript ID ijms-1177509

Type Review

Number of Pages 18

Title: Links between immune cells from the periphery and the brain in the pathogenesis of epilepsy: A narrative review

To the Reviewer 2

We appreciate the time and effort you have dedicated to providing this insightful feedback.  All suggested revisions, as well as additional ones to improve the language of the manuscript, are indicated in underline and yellow text. We sincerely hope that, with these revisions, our manuscript will be suitable for publication in your esteemed journal.

Reviewer: 2
Comments to the authors

Thank you very much for your comment. We made the following correction (line 146).

Pilocarpine is a muscarinic receptor agonist, which is often used to induce convulsions, similar to KA.

to

Pilocarpine is a cholinergic agonist used as a tool to induce seizures.
